



# Static-Gradient NMR imaging for Depth-Resolved Molecular Diffusion in Amorphous Regions in Semicrystalline PTFE Film

Natsuki Kawabata[1], Naoki Asakawa[1], and Teruo Kanki[2]

[1]Material Science Program, Division of Materials and Environments, Graduate School of Science and Technology, Gunma University, Kiryu, Gunma 3768515, Japan
[2]Institute of Scientific and Industrial Research, Osaka University, Ibaraki, Osaka 567-0047, Japan

**Correspondence:** Naoki Asakawa (asakawa@gunma-u.ac.jp)

**Abstract.** Understanding spatially heterogeneous molecular diffusion in semicrystalline polymers is critical for elucidating interfacial dynamics in soft materials. This study employs static-gradient nuclear magnetic resonance (NMR) imaging to capture depth-resolved translational motion of polymer chains in a polytetrafluoroethylene (PTFE) film. By focusing on spin–spin relaxation behavior in amorphous regions near crystalline lamellae, we identify multiple diffusion regimes consistent with

Bloch–Torrey analysis. The results reveal that molecular mobility at the substrate interface is significantly constrained, likely due to interfacial pinning, while the air-side surface shows signs of enhanced mobility. Our findings highlight the utility of static-field-gradient NMR for probing nanoscale dynamical heterogeneity in semicrystalline systems.

## 1 Introduction

Heterogeneity in the dynamical behavior of polymer films—manifested as distinct molecular dynamics at the air-facing surface,

within the bulk, and at the substrate interface—profoundly impacts the physical properties of these materials, particularly their dynamic viscoelasticity(Keddie et al. (1994); De Gennes (2000); Fukao and Miyamoto (2000); Merabia et al. (2004); Roth and Dutcher (2005); Fakhraai and Forrest (2008); Ediger and Forrest (2014); Inoue and Kanaya (2013)). Over the past decades, considerable efforts have been devoted to clarifying how interfacial and surface regions contribute to phenomena such as the glass transition, highlighting the critical role of site-specific molecular mobility. To further unravel these complex

dynamics, analytical techniques that can provide spatially resolved information about molecular motion in polymer thin films are indispensable.

Nuclear magnetic resonance (NMR) spectroscopy offers a powerful and noninvasive means of probing molecular structure and dynamics across a broad range of materials(Westbrook and Talbot (2018); Blumich (2000); Blümich et al. (2008); Blümich (2019)). Among NMR-based approaches, pulsed field gradient (PFG) methods have been widely adopted because they enable

both high-resolution spectroscopy and magnetic resonance imaging (MRI)(Callaghan (1984); Price (1997); Mansfield et al. (1976); Mansfield (1977)). Nevertheless, while PFG methods are well suited for liquid systems, the gradient strengths achievable are often insufficient to study solid-state specimens or to capture diffusion processes with extremely small diffusion coefficients(Kimmich et al. (1991); Chang et al. (1994, 1996); Ailion (1999)). This limitation underscores the need for alternative approaches that are specifically tailored for solid materials.



To overcome these challenges, methodologies utilizing static magnetic field gradients (SFG) have been developed as compelling alternatives(Chang et al. (1994, 1996); Ailion (1999)). Unlike PFG, which applies gradients in a pulsed manner, the SFG approach relies on a continuously imposed gradient. Although early implementations typically exploited fringe fields from electromagnets and were limited by thermal and power constraints, subsequent advances demonstrated that sufficiently strong gradients could be realized using superconducting magnets(Kimmich et al. (1991)). As a result, SFG has enabled suc-

cessful quantification of self-diffusion coefficients in solids and has stimulated the design of a wide variety of cost-effective platforms, such as the superconducting fringe field (SFF) technique(Kimmich et al. (1991)), stray-field imaging (STRAFI) employing GARField magnets(Dias et al. (2003)), anti-Helmholtz superconducting magnets(Chang et al. (1994, 1996)), NMR MOUSE(Eidmann et al. (1996)), single-sided NMR systems(Blümich et al. (2008)), bulk high-temperature superconducting magnet-based methods8Takahashi et al. (2022)), Halbach-array NMR sensors(Raich and Blümler (2004); Doğan et al. (2009);

Tayler and Sakellariou (2017); Chang et al. (2006)), and compact ferromagnet-based MRI systems(Asakawa and Obata (2012)). These developments clearly demonstrate the potential of SFG-based MRI to complement or even replace conventional PFG methods, particularly in the study of solid materials.

Despite these advances, PFG methods still face intrinsic drawbacks that limit their utility for solid-state applications. The presence of electrically conductive constituents within the sample can induce substantial eddy currents, potentially resulting in

sample degradation. Additionally, the abrupt shifts in the electromagnetic environment triggered by gradient pulses can lead to prolonged dead time in the magnetic resonance signal(Chapman et al. (1957); Gibbs and Johnson Jr (1991); Price (1998)). Moreover, conventional high-frequency NMR systems reliant on superconducting magnets necessitate stringent operational infrastructure, including large-scale facilities, cryogenic cooling, and vacuum environments, all of which contribute to substantial financial and logistical burdens. Consequently, there remains a pressing need for a simple, cost-effective, and versatile MRI

method that can provide spatially resolved information on molecular dynamics in solid systems.

A further complication is that, in many NMR systems, the static magnetic field governing resonance conditions and the magnetic field gradient used for imaging or diffusion measurements are not independently controllable. This interdependence complicates the execution of NMR measurements across different frequencies while preserving consistent spatial resolution. Nevertheless, there is a growing demand for the spatially resolved acquisition of spectral density functions—namely, the local

spectral density—which facilitates advanced imaging modalities. Bridging this gap requires the development of novel MRI methodologies that can combine simplicity, tunability, and sensitivity to solid-state molecular motion.

In this context, the present study introduces a novel, non-destructive, and facile molecular dynamics imaging technique. This approach employs a locally generated magnetic field gradient from a needle-shaped ferromagnetic material developed in-house. Using this technique, depth-resolved spin–spin relaxation rate ($R_2$) imaging of a polymer film sample is performed,

enabling direct comparison of molecular dynamics at the film surface and near the substrate interface.

Our previous study measured the variable-frequency spin–lattice relaxation rate ($R_1$) to determine the spectral density function associated with local molecular motion(Kawabata et al. (2024)). Although the thicknesses of the surface and interface estimated from $R_1$-variation were too large due to $^{19}$F-$^{19}$F spin diffusion, the disparity between the surface/interface and interior of the PTFE film was confirmed. However, the $R_1$ measurements indicated no discernible differences between the $R_1$





values at the air-side surface of the polymer bulk film and near the polymer–substrate interface. This finding diverges from the widely recognized behavior observed during the glass transition of conventional thin glass-forming polymer films, wherein $T_g$ typically decreases at the air-exposed surface(Forrest et al. (1996); Forrest and Mattsson (2000); Mattsson et al. (2000); Dalnoki-Veress et al. (2001); Park and McKenna (2000)) and increases in the vicinity of the polymer–substrate interface(Lin et al. (1999); Fryer et al. (2000); Tanaka et al. (2006); Nguyen et al. (2019)). This discrepancy may stem from the fact that $R_1$

reflects solely molecular rotational dynamics, rather than translational diffusion.

Thus, to complement our previous work, we examined the influence of translational diffusion on spin–spin relaxation using the $R_2$-dispersion method(Yu (1993)) by systematically varying the echo time of the spin-echo method. It should be noted here that the PTFE sample is a highly crystalline polymer with a degree of crystallinity of about 90%(Kawabata et al. (2024)). In the CPMG spin echo method used in this study, the NMR signals from the crystalline parts are thought to be removed from

70 the obtained spin echo due to the fast spin-spin relaxation decay caused by the large $^{19}$F-$^{19}$F magnetic dipole coupling. The obtained NMR signals are thought to be mainly from the amorphous parts present on the crystal grain surface. The diffusion of such amorphous molecules is thought to be restricted by the surrounding molecules and crystalline parts, even though they are in a rubber state. Therefore, as will be described later, the diffusion in this study is thought to be a diffusion behavior in a compartmented confined space.

## 2 Methods

### 2.1 Principle of depth profiling

The magnetic-field gradient experienced by the sample arises from the combined influence of the static magnetic field and its gradient along the thickness of the polymer film. This gradient is induced by a needle-shaped ferromagnetic material and the uniform external static magnetic field generated by the electromagnet. The interplay between static magnetic field gradient and

80 external static field effectively compensates for the insufficient magnetic field strength to generate NMR signals.

Consequently, a spatial region at which the magnetic field intensity reaches the required threshold for the onset of nuclear magnetic resonance (NMR), referred to as the excited volume, is established (see Fig. 1(a)).

Within this framework, the spin density or molecular dynamics at a specific location within the sample can be characterized by analyzing the NMR signal arising from the intersection volume between the excited volume and the sample. The morphology

of the excited volume, which is shaped by the presence of a needle-shaped ferromagnetic material, is anticipated to adopt a concave geometry, as illustrated in Fig. 1(a)(Degen et al. (2009); Chao et al. (2004)). The excited volume can be displaced vertically by modulating the strength of the external static magnetic field produced by the electromagnet, as shown in Fig. 1(a) and (b). Systematic variation of the external static magnetic field enabled the acquisition of a depth-resolved profile of the sample.





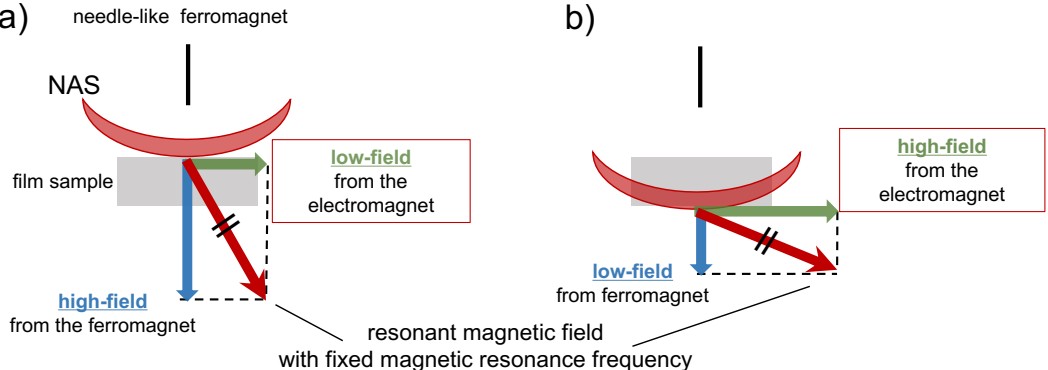

**Figure 1.** The combined effect of the magnetic field generated by the needle-like ferromagnet, $B_{\mathrm{N}d}$, and the static magnetic field from the electromagnet, $B_{\mathrm{e}}$ gives rise to the resonant field, namely the excited volume. The morphology of the excited volume, shaped by the needle-like ferromagnet, exhibits a meniscus-lens-like structure. All NMR measurements are conducted under a fixed resonance frequency, with a stepwise variation of $B_{\mathrm{e}}$. By adjusting the strength of the static magnetic field from the electromagnet, the excited volume can be vertically displaced relative to the sample plane. When the static magnetic field from the electromagnet is weak, the magnetic field generated by the needle-like ferromagnet dominates, causing the excited volume to shift upward, closer to the needle (a). Conversely, when the static magnetic field is strong, a weaker contribution from the needle-like ferromagnet suffices, leading to a downward displacement of the excited volume, away from the needle (b).

## 2.2 Experimental setup

The experimental setup for this methodology is illustrated in Figure 2. A spherical neodymium magnet (8 mm diameter sourced from TRUSCO Nakayama Corporation) and an iron needle (1 mm diameter with a tip diameter of 0.2 mm) were affixed to the aluminum jig. By bringing them in direct contact, the iron needle is magnetized and transformed into a ferromagnetic needle that functions as a localized ferromagnetic material. The needle was positioned between the poles of a water-cooled electro-magnet, serving simultaneously as a static magnetic field source and static magnetic field gradient generator. For the detailed specifications of the apparatus, please refer to our previously published work(Kawabata and Asakawa (2024); Kawabata et al. (2024)).

## 3 Results and Discussion

### 3.1 Depth-direction one-dimensional spin–spin relaxation rate imaging of single layer of polymer film

To investigate the effect of translational molecular diffusion at the film surface and the polymer-glass interface, we performed one-dimensional imaging of the spin-spin relaxation rate ($R_2$) of $^{19}$F nuclei along the depth of a 2-mm-thick PTFE film immo-bilized on a glass substrate using epoxy resin. From the blurred image of the sample, which is determined from magnetization





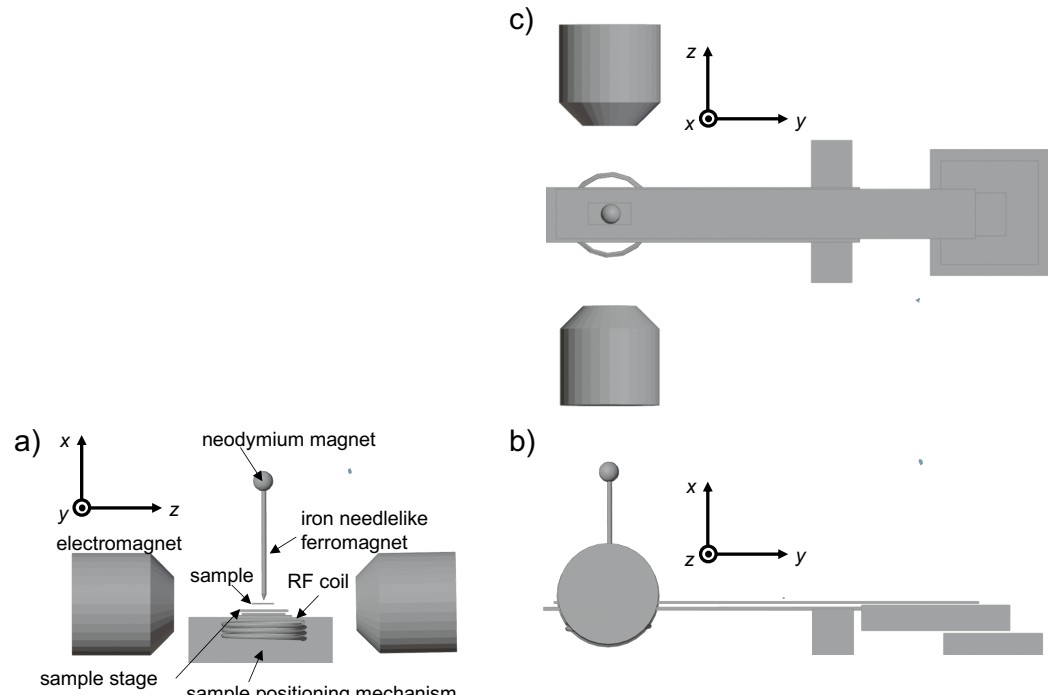

**Figure 2.** Schematic illustration of an MRI probe. Three-dimensional views from the $y$-axis (a), $x$-axis (b), and $z$-axis (c) are shown, respectively. A small spherical neodymium magnet magnetizes the paramagnetic needle, converting it into a needle-like ferromagnet. Two-dimensional mechanical scanning stage, is shown (only the function for sample positioning is used and the scanning function is not used in this work).

intensity, this method achieves spatial resolution on the order of sub-millimeter scale, which enables observation of mesoscopic heterogeneities.

The Carr–Purcell Meiboom–Gill (CPMG) sequence was employed for the MRI measurements, and the dependence of $R_2$ on translational diffusion was examined by varying the echo time, $\tau$, as follows: Depth-resolved one-dimensional imaging was achieved by stepwise modulation of the static magnetic field strength using a normal-conducting electromagnet. At each magnetic field point, 256 signal accumulations were acquired at a resonance frequency of 29.750000 MHz. The decay plots and fitting curves for the CPMG measurements are shown in Appendix A. Furthermore, the influence of the intrinsic $R_2$ on

the experimentally obtained $R_2$ was negligibly small(see Appendix B). However, in the $R_2$-dispersion method employed in this study, accurately determining the diffusion coefficient is challenging due to the ambiguous nature of the local magnetic field gradient within the sample. Consequently, it should be emphasized that the diffusion analysis presented here is qualitative in nature. The resulting $R_2$ imaging data are presented in Fig. 3, where: $M_{inf}$ represents the NMR signal intensity. In this imaging approach, the horizontal axis corresponds to the static magnetic field strength generated by the electromagnet, with





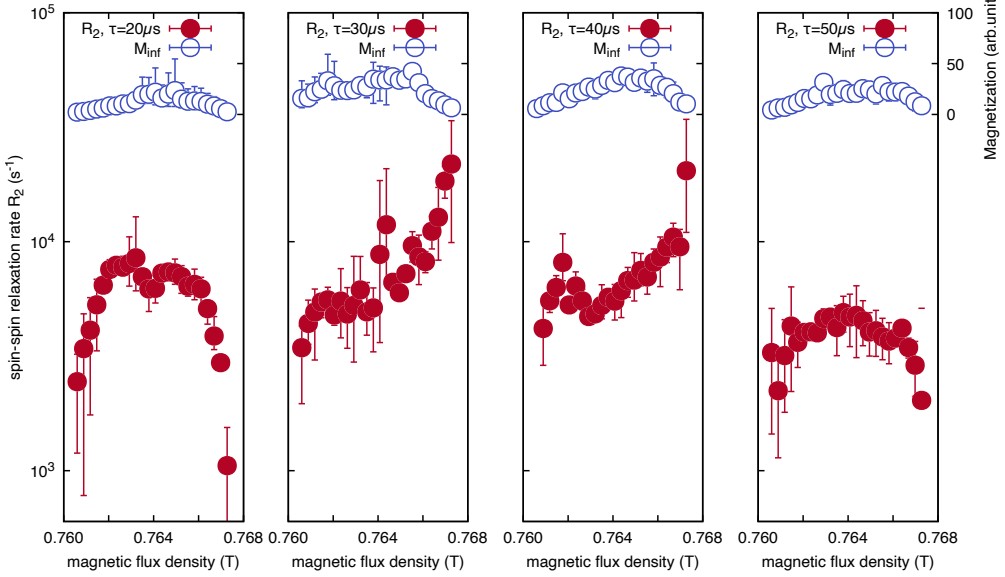

**Figure 3.** Magnetic field dependence (corresponding to depth dependence)of the $T_2$ relaxation rate($R_2$) measured using the CPMG method at different echo times for a PTFE film adhered on a soda-lime glass substrate with epoxy resin. $R_2$ rate showed minimal contrast at the PTFE film surface ($B_e < 0.762$ T), whereas it displayed remarkable changes at the substrate interface ($B_e > 0.766$ T).

an increasing field strength corresponding to deeper regions within the sample. Thus, the region of lower static magnetic-field strength represents the air-facing surface of the film, whereas the region of higher static magnetic-field strength corresponds to the interface with the glass substrate. As shown in Fig.3, no variations present in $R_2$ near the air-side surface of the film, ($B_e < 0.762$ T) for different $\tau$ values. A similar trend was observed in the interior of the film. However, near the glass-side interface of the film ($B_e > 0.766$ T), complex behavior in $R_2$ emerged. Specifically, as $\tau$ increases $R_2$ initially exhibits an

increase ($\tau = 30\,\mu$s) before subsequently decreasing for longer $\tau$ values ($\tau = 40$ and $50\mu$s). The contrasting behavior of $R_2$ at the air- and glass-side interfaces of the film could be attributed to different molecular interactions. At the air interface, PTFE molecules behave as free ends owing to the surface energy effects of interactions with adjacent PTFE molecules. Conversely, near the glass substrate interface, the pinning effect induced by interactions between epoxy resin and PTFE molecules is presumed to constrain translational diffusion, thereby influencing the observed relaxation dynamics.

Based on the experimental findings presented above, we hypothesize that the variation in the dependence of $R_2$ on increasing $\tau$ arises from differences in the translational diffusion effects. This is modulated by the distinct interfacial environments of the PTFE film, such as air or the glass substrate.





## 3.2 Three Diffusional Regimes and Regime Transitions

Figure 4(a) illustrates the results of numerical simulations using the Bloch–Torrey equation (see Appendix C). The variation

in diffusion regime with the dimensionless diffusion coefficient $\tilde{D}(= D\tau/L_s^2)$ and position $x$ of the nuclear spin in real space

is represented on the xy-axes of the plot. The contribution of relaxation exponent of spin–spin relaxation rate $R_2$ due to

diffusion is depicted along the $z$ axis. When the echo time $\tau$ is short, and diffusion is not in progress; the molecule is far

from the diffusion barrier and remains within the short-time regime. Here, the molecule is free to diffuse throughout the space.

As molecular diffusion advances and the molecule approaches the diffusion barrier, it transitions into a localization regime

where diffusion is constrained by the barrier. As the diffusion continues, the molecule enters the motional averaging regime,

undergoing multiple round trips between the diffusion barriers.

In this regime, molecular motion undergoes averaging, such that the system appears stationary and diffusion is not observed.

Given the difficulty of observing the three-dimensional curve in Figure 4(a), the sum of the magnetization versus position in

real space is plotted with respect to $\tilde{D}$ in Figure 4(b). The figure shows the presence of three distinct regimes.

## 140   3.3 Distinction between Surface and Interface

Building on the aforementioned observations, we examine the behavior of $R_2$. A consistent trend was observed across all

echo times. $\tau$ conditions used in the experiment were as follows: $R_2$ was notably smaller on the air-side surface of the PTFE

film ($B_e < 0.762$ T) than in the interior of the film. This behavior of $R_2$ arises from its intrinsic modification instead of the

diffusion process. A reduction in $R_2$ on the air-side surface was observed irrespective of the variation in $\tau$. Results from

variable frequency $R_1$ measurements(Kawabata et al. (2024)) indicated that, within the resonant frequency range (29.75 MHz)

employed for the $R_2$ measurements, $R_1$ value near the surface was 1.5–2 orders of magnitude larger than the $R_1$ of the

film. Hence, the contributions of the $R_1$ to $R_2$; that is, the effect of the secular term, was considered negligible. Therefore, we

hypothesize that a reduction in $R_2$ observed at the film surface can be attributed to differences in the zero-frequency component

of the spectral density function, $J(\omega = 0)$. Based on the established variation in the spectral density of molecular motion near

the film surface or substrate interface(Kawabata et al. (2024)), PTFE molecules are postulated to exhibit enhanced molecular

motion, particularly the rotational motion at zero or low frequencies below several hundred kilohertz, compared to the bulk

of the film. It should again be emphasized that the observed thicknesses of the surface and interfacial regions are significantly

greater than the typical values – on the order of several tens of nanometers – commonly reported for surfaces and interfaces

in nanometer-scale thin films(Forrest et al. (1996); Forrest and Mattsson (2000); Mattsson et al. (2000); Dalnoki-Veress et al.

(2001); Park and McKenna (2000); Lin et al. (1999); Fryer et al. (2000); Tanaka et al. (2006); Nguyen et al. (2019)). This

discrepancy is attributable to the spin diffusion effect of $^{19}$F-$^{19}$F interactions on $R_2$.

### 3.3.1 Behavior of the air surface/interior of the polymeric film

We now examine the impact of variations in the echo time $\tau$ on the air-side surface and the interior of the PTFE film. The results

demonstrate that the value of $R_2$ remains largely unchanged even with the echo time $\tau$ gradually increases. By comparing this





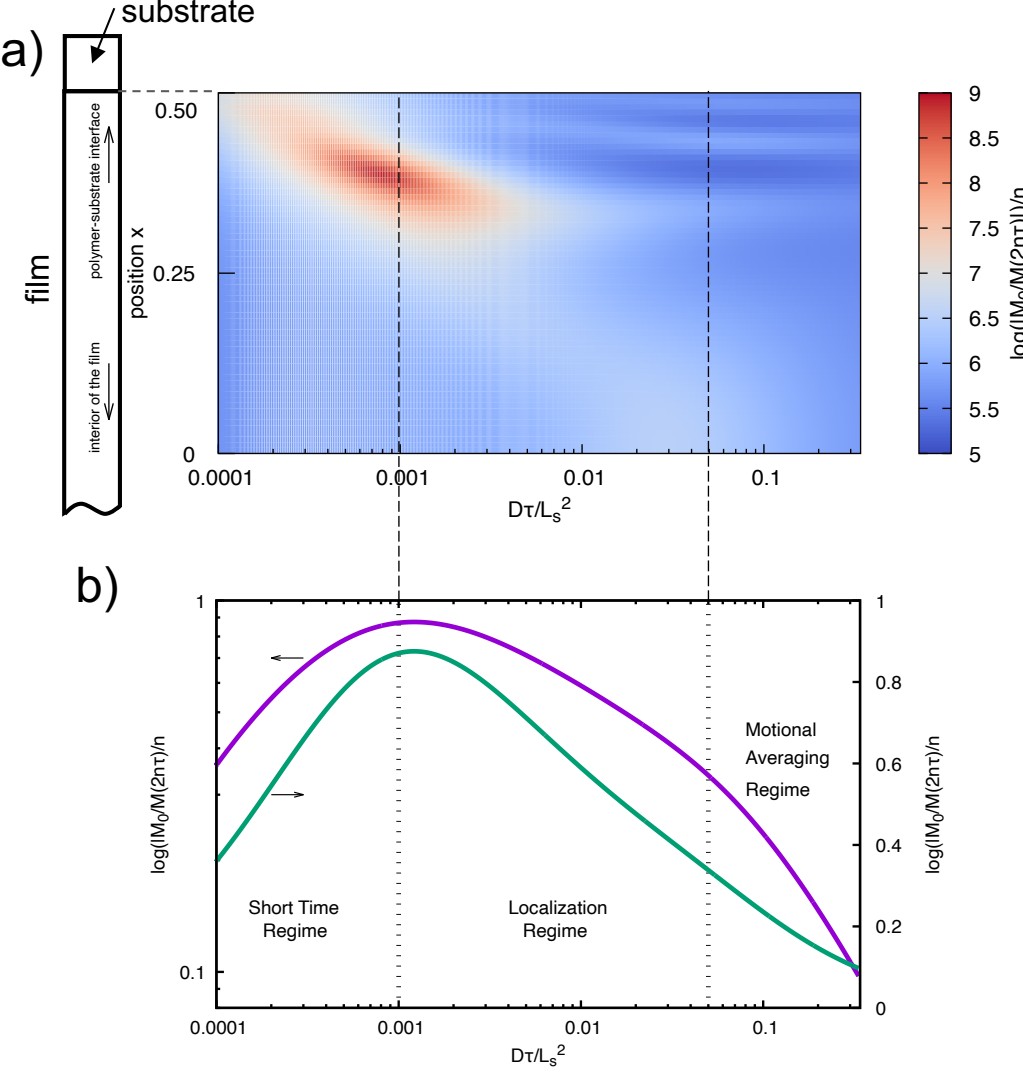

**Figure 4.** Simulation of the relaxation exponent of the first echo signal of CPMG experiment using the Bloch–Torrey equation for the diffusion of nuclear spins in a compartmentalized environment. Panel (a) presents the relaxation exponent for the CPMG echo intensity ($M(2n\tau)$; $n=1$) as a function of the dimensionless diffusion coefficient (or dimensionless echo time) and position in a sample. The position is defined relative to the midpoint between diffusion barriers of the compartment, with a magnetic field gradient symmetrically distributed about the origin, having the form $g(x) = x^4$. The figure presents data for only half of the compartment. The simulation models a polymer film confined between two substrates in order to apply periodic boundary conditions. In practice, however, only one side of the polymer film adheres to the substrate, while the opposite surface is exposed to air. Thus, the film possesses an asymmetric structure along the depth direction, with one interface in contact with the substrate and the other with air. Consequently, only the results corresponding to the substrate–film interface are displayed. The dimensionless diffusion coefficient is expressed as $D\tau/L_s^2$, where $L_s$ represents the distance between the diffusion barriers of the compartment, $D$ is the diffusion coefficient, and $\tau$ is the CPMG echo time. Panel (b) illustrates the total observed magnetization within the compartment as a function of the dimensionless diffusion coefficient, highlighting the three regimes.





behavior with Figure 4(a), the system is in the localization regime, where changes in $\tau$ exert a minimal influence on $R_2$. This suggests that the air-side surface acts as a free end, where the dynamics of individual polymer chains are not entirely random, and that these chains function as a diffusion barrier in the direction normal to the film surface because of their interactions with adjacent polymer chains. Moreover, since the $R_2$ value of the PTFE molecules within the film was independent of the variations in $\tau$, the behavior observed on the film surface can be attributed to the film interior. However, within the film, unlike the one-dimensional diffusion barrier perpendicular to the film surface, localization is presumed to arise from collisions with the three-dimensional barriers formed by the surrounding PTFE molecules.

### 3.3.2 Behavior of the interface between the polymeric film and the substrate

We now focus on the region near the interface between the PTFE film and the glass substrate. Near the glass-side interface of the PTFE film ($B_e > 0.766$ T), $R_2$ exhibits notable variation when $\tau$ shifts from 20 $\mu$s to 30 $\mu$s. However, as $\tau$ reaches 40 $\mu$s, the value of $R_2$ begins to decline, and by the time, $\tau$ reaches 50 $\mu$s, $R_2$ decreases precipitously. This behavior can be attributed to the system transitioning between the localization and motional averaging regimes, triggered by the change in $\tau$ as shown in Fig. 4(a).

### 3.4 Pinning effect at interaface

The diffusion behavior of PTFE molecules near the PTFE film interface differed from that in the film interior or near the film surface, owing to the influence of the epoxy resin. This disparity is because of employing the same sample for all the experiments conducted at 298 K, under the assumption that the diffusion coefficient $D$ remains constant under isothermal conditions. Here, $D$ represents the diffusion coefficient as the statistical average of random motion, akin to Brownian motion. In other words, $D$ corresponds to the diffusion coefficient in the Fokker–Planck equation, which is used when the Langevin equation for a single molecule that accounts for random forces due to thermal fluctuations at a given temperature is extended to a molecular ensemble. Since $D$ and $\tau$ are constants, in $\tilde{D}(= D\tau/L_g^2)$, only $L_g$ denotes a variable. Here, the distance between diffusion barriers $L_s$ is replaced by the effective diffusion barrier distance, which is denoted as the spin packet length $L_g$ (see Appendix D for more details).

Specifically, when $\tau = 20\mu$s was held constant, and a regime transition was observed in the depth direction of the polymer film, as depicted in the plot in Figure 4c). This transition occurred between the localization and motional averaging regimes. Since $D$ and $\tau$ are constants, and the variations in $\tilde{D}$ is attributed to the changes in $L_g$, the transition is triggered by a reduction in the spin-packet length $L_g$.

Near the substrate interface, significant changes in $\tau$ causes $R_2$ to transition from increasing to decreasing. Thus, a reduction in $L_g$ and the increase in $\tau$ leads to an increase in the $\tilde{D}$, which results in regime transition. Here, we explored this phenomenon. On the airside surface of the PTFE film, despite the variations in $\tau$ during CPMG measurements, the value of $R_2$ exhibited a minimal change, indicating that the molecules on the air-side surface of the film were in the localization regime. In other words, the PTFE molecules on the air-side surface exhibited behavior consistent with restricted diffusion. However, when considered in conjunction with the experimental results for $R_1$ reported previously(Kawabata et al. (2024)), the effect of translational



diffusion was found to be equivalent to that observed in air-side surface and the interior of the PTFE film, with the observed difference in $R_2$ arising from variations in the spectral density function, $J(\omega = 0)$, which is attributed to rotational motion.

This conclusion differs from the well-known results of translational diffusion of glassy polymers near the air-side surface, as revealed by fluorescence lifetime experiments and coarse-grained molecular dynamics simulations(Tanaka et al. (2009)), in which a polymeric thin film shows a decrease in the glass transition temperature at the surface. This discrepancy may be attributed to the fact that our $R_2$-dispersion experiments were carried out under rubbery-state conditions, at temperatures significantly higher than $T_g$.

On the other hand, the observed relationship for the glass–substrate interface can be explained as follows: The spin packet length $L_g$ diminishes near the glass–substrate interface compared with that within the bulk of the film. There are two potential explanations for the reduction in $L_g$. The first explanation involves an increase in the strength of the local magnetic field gradient owing to the contrast in magnetic susceptibility at the interface between the PTFE film and epoxy resin on the glass substrate. However, in this case, the effect of diffusion on $R_2$ increases monotonically. This is because when the magnetic field

gradient strength is simply enhanced, the transition point between the localization regime and the motional averaging regime shifts to a larger $\tilde{D}$, making traversing the transition point by merely increasing $\tau$ challenging. In this study, the transition from an increase to a decrease in $R_2$ owing to an increase in $\tau$, that is, a regime transition, was observed, suggesting that the contribution from changes in the local magnetic field gradient is relatively minor compared with the contribution from the increase in $\tau$. That is, a reduction in the $L_g$ can be attributed to the second potential explanation, the diffusion of PTFE

molecules. We speculate that the most likely mechanism for the phenomenon is the pinning effect of PTFE chains at the interface between the PTFE film and glass substrate. This pinning effect led to a reduction in $L_s$, which subsequently results in a decrease in the $L_g$.

To provide a unified interpretation of this phenomenon, we applied the Bloch–Torrey equation, which integrates the effects of diffusion into the Bloch equation and the CPMG spin-echo experiment, followed by analyzing the diffusion regime transition

in one-dimensional MRI. As illustrated in Figure 4a), compared with the interior of the film, which is distant from the diffusion barrier, the regime transition occurs at a smaller $\tilde{D}$ near the diffusion barrier where the magnetic field gradient is substantial.

## 4  Conclusions

As described above, NMR $R_2$ imaging of the depth profile of a polymer film utilizing a static magnetic field gradient generated by a needle-shaped ferromagnetic material revealed distinct variations in the diffusion behavior near the air-side surface of a

PTFE polymer film compared to the film interior and substrate-side interface. The methodology developed in this study offers a novel approach for imaging heterogeneous materials and dynamic imaging of molecular processes.

*Data availability.* The datasets generated and analyzed during the current study and software code for solvation of Bloch-Torrey equation are available from the corresponding author on reasonable request.



## Appendix A: $R_2$ analyses from CPMG measurements

For CPMG measurements, the plots of decay data for each excited volume as a function of $2n\tau$ are shown in Figure. A1. Here, we discuss regarding the determination of $R_2$. The one-dimensional images reconstructed from the acquired signal intensities appear as blurred images, rather than reflecting the true rectangular geometry of the sample, resulting in broadened profiles. These blurred images arise from the convolution of the true sample geometry with the point spread function, which itself is determined by the spatial dependence of the magnetic field strength within the sample. Consequently, the data points in adjacent regions of the image do not originate from spatially independent excited volumes, but instead represent correlated points. While this characteristic constitutes a limitation of the present method, it also suggests the possibility that, in the future, inverse problem approaches to convolution–such as the Landweber iteration method–may enable imaging with significantly enhanced spatial resolution.





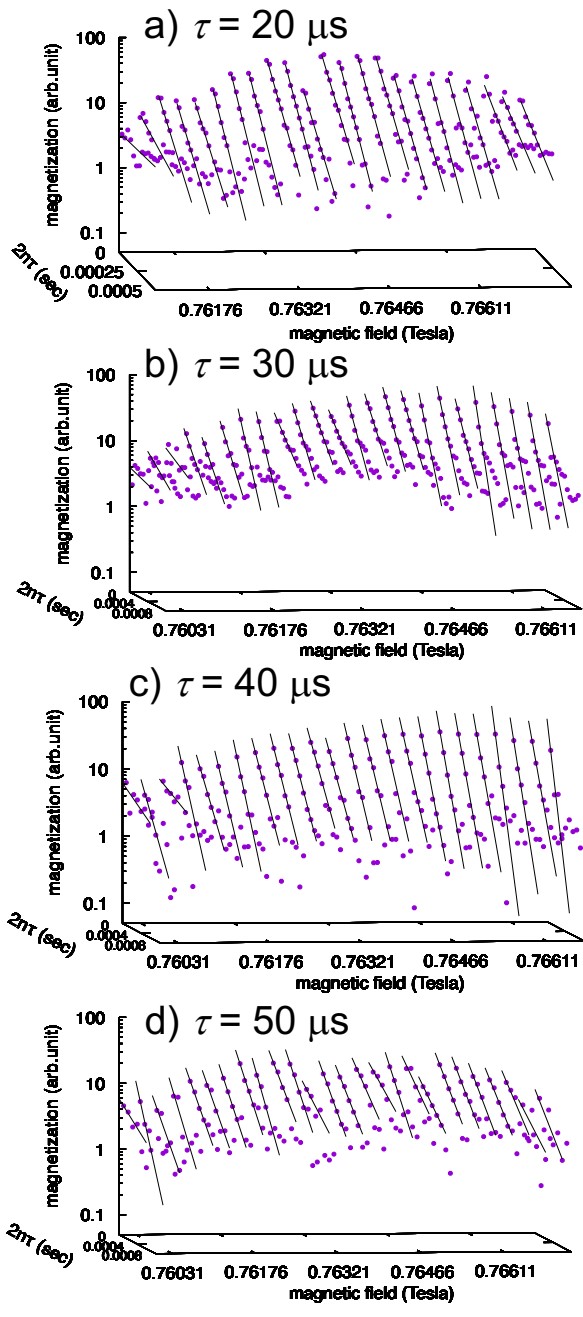

**Figure A1.** One-dimensional depth profiling of the CPMG decays for the PTFE film. The decay was fit using $M(2n\tau) = M_{inf} \exp(-R_2\tau)$ for the early time period of each decays.





## Appendix B: Intrinsic $R_2$

Distinguishing between the intrinsic $R_2$ and the contribution arising from diffusion is crucial for accurately evaluating the influence of diffusion on $R_2$. However, when the intrinsic $R_2$ lies within the short-time regime, namely the free-diffusion regime, it can be determined by extrapolating the $R_2$ *vs.* $\tau$ plot to $\tau \to 0$. Similarly, when the diffusion process resides in the motional averaging regime, the intrinsic $R_2$ can also be estimated. In contrast to the short-time regime, however, extrapolation concerning $\tau$ is not possible; instead, if the plot exhibits a value asymptotically approached with increasing $\tau$, this value can be

regarded as the intrinsic $R_2$. By contrast, when the system falls into the localization regime, these methods cannot be applied. Furthermore, as demonstrated in the present study, in cases where regime transitions occur with variations in $\tau$, determination of the intrinsic $R_2$ may not be straightforward. Therefore, we attempted to estimate the intrinsic $R_2$ from the $\tau$ dependence of $R_2$ in the motional averaging regime. However, since the $\tau$ dependence did not exhibit a clear asymptotic value, the intrinsic $R_2$ could not be determined. Nevertheless, from the peak-like feature observed in the $\tau$ dependence of $R_2$ near the interface

with the substrate, it can be inferred that $R_2$ already decreases to approximately $10^2$ s$^{-1}$ or less at an echo time of $\tau = 100$ $\mu$s. This suggests that the maximum intrinsic $R_2$ is on the order of $10^2$ s$^{-1}$ and is likely smaller.

According to the Solomon-Boembergen theory( Solomon (1955)), using a correlation time of 1 $\mu$s estimated from $R_1$ and the second moment of the $^{19}$F-$^{19}$F magnetic dipolar coupling, $M_2 = (2\,\pi \times 36.568$ kHz$)^2 = 5.27 \times 10^{10}$ s$^{-2}$, the relaxation rate can be estimated as $R_2 \approx M_2\,\tau_c \approx 1 \times 10^4$ s$^{-1}$. This value exceeds the experimentally obtained $R_2$, indicating that

employing the correlation time corresponding to crystalline domains is not appropriate. In the CPMG method, signals from crystalline regions are attenuated by rapid $R_2$ relaxation during the echo intervals; therefore, the observed signals in this study are considered to originate solely from the amorphous regions. By adopting the reported correlation time of $\tau_c = 4.3 \times 10^{-14}$ s for the amorphous phase, $R_2$ is estimated as $2.3 \times 10^{-3}$ s$^{-1}$. Assuming that the intrinsic $R_2$ contributes within the range of $10^{-3}$-$10^2$ s$^{-1}$ to the experimentally observed $R_2$, this contribution is negligibly small. Consequently, in this work, the

experimentally obtained $R_2$ values were plotted without subtracting the intrinsic $R_2$ component.

## Appendix C: Numerical simulation

Magnetization decay in a spin-echo experiment subjected to a static magnetic field gradient can be simulated by numerically solving the Bloch–Torrey equation(Torrey (1956); Axelrod and Sen (2001); Asakawa et al. (2005); Asakawa and Obata (2012)). This section presents a solution to the Bloch–Torrey equation within a compartmentalized diffusion environment. In

this context, the compartments emulate the cage effect imposed by the surrounding molecular segments of the polymer chains, a phenomenon frequently considered in the study of glass transition behavior(Doliwa and Heuer (1998)) or amorphous polymer chains between crystalline regions.

The Bloch–Torrey equation is given by:

$$\frac{\partial M}{\partial t} = \tilde{D}\frac{\partial^2 M}{\partial \tilde{x}^2} - i\tilde{\gamma}G(\tilde{x})M, \tag{C1}$$





where the tilde denotes the dimensionless parameters, allowing the problem to be treated in a generalized framework. The contributions of the longitudinal relaxation ($R_1$) and intrinsic transverse relaxation ($R_2$) in the original Bloch–Torrey equation were neglected in this calculation. Although these effects must be accounted for in practical applications, it is found that the effects are negligible in our case.

The dimensionless diffusion coefficient $\tilde{D}$, the dimensionless gyromagnetic ratio $\tilde{\gamma}$, and the dimensionless one-dimensional coordinates $\tilde{x}$ is defined as

$$\tilde{D} = \frac{D\tau}{L_s}, \tag{C2}$$
$$\tilde{\gamma} = \tau g\gamma L_s, \tag{C3}$$
$$\tilde{x} = \frac{x}{L_s}, \tag{C4}$$

where $D$, $L_s$, $\gamma$, and $x$ represent the diffusion coefficient, the characteristic length of the aforementioned compartment, gyromagnetic raito of a noticed nuclear spin, and the spatial position of the nucleus within the molecule, respectively.

Because the temporal sequence during echo time $\tau$ is irrelevant in transverse relaxation measurements , $\tau$ can be considered as a single computational step. Time evolution operator for spin-echo experiment over echo time $\tau$ is given by:

$$U_+ = e^{(\hat{W}-i\tilde{\gamma}\hat{B})\tau}. \tag{C5}$$

Thus, the propagator of a two-pulse Hahn echo(Hahn (1950)) can be expressed as follows:

$$U_{2\tau} = U_-U_+ = (U_+)^*U_+. \tag{C6}$$

Similarly, the propagator of the second echo in the Carr–Purcell Meiboom–Gill (CPMG) sequence(Carr and Purcell (1954); Meiboom and Gill (1958)) is given by

$$U_{4\tau} = U_+U_-U_-U_+. \tag{C7}$$

Extending this to the $n$th echo in the CPMG sequence, that is, the propagator, is

$$U_{2n\tau} = U_{2\tau}^{mod(n,2)}U_{4\tau}^{floor(n,2)}. \tag{C8}$$

Consequently, the $k$th Fourier component of the magnetization at time $t = 2n\tau$ is computed as follows:

$$M(k,t) = U_{2n\tau}M(k,0). \tag{C9}$$

Initial magnetization $M(k,0)$ is determined using the following initial conditions: ($\phi(\tilde{x},0)$):

$$\sum_{k=0}^{\infty} M(k,0)\psi_k(\tilde{x}) \tag{C10}$$
$$= \sum_{k=0}^{\infty} M(k,0)\cos\left(\pi k\left(\tilde{x}+\frac{1}{2}\right)\right) \tag{C11}$$
$$= \phi(\tilde{x}). \tag{C12}$$





The Fourier component $M(,k,0)$ of the initial magnetization can be obtained using the inverse Fourier transform of $\phi(\tilde{x})$ and For instance, for a homogeneous spin density, where $\phi(\tilde{x}) = 1$ (constant), all the Fourier components vanish, except for $k = 0$ because the inverse Fourier transform yields a delta function. Substituting the obtained $M(k,0)$ into Eq. C9, the $k$-th magnetization, $M(k,t)$ at time $t(= 2n\tau)$ can be determined. Finally, the magnetization in real space $M(x,t)$ is obtained using the inverse spatial Fourier transform of $M(k,t)$:

$$M(x,t) = \mathcal{F}^{-1}[M(k,t)]. \tag{C13}$$

In Panel a of Fig.4 in the main text, it is not straightforward to clearly identify the three diffusion regimes; therefore, in Panel b, we presented the vertical axis as the spatial summation of magnetization over all positions $x$. By stating that the results were "mapped to real space," we meant that, after solving the Bloch-Torrey equation in Fourier space, we applied the inverse Fourier transform to recover $M(x,t)$ in real space. The plot in Panel b of Fig.4 in thre main text further shows $\sum_x M(x,t)$ as a function of the dimensionless echo time, representing the total magnetization integrated over all positions. The purpose of Panel b was thus to make the visualization of the three diffusion regimes more accessible. Panel a, on the other hand, demonstrates that in one-dimensional imaging the echo times at which regime transitions occur differ between regions near the substrate interface and those deeper within the film.

A Fourier transform approach was employed to solve the Bloch–Torrey equation, which implicitly imposes periodic boundary conditions. In the case of a PTFE film sandwiched between two substrates, the interactions at both film surfaces with the substrates create a symmetric structure along the film thickness, making the direct application of periodic boundary conditions appropriate. However, in the present study, the polymer film exhibits an asymmetric structure, with one surface exposed to air and the other interfacing with the substrate. Therefore, we used only the simulation results corresponding to the half-depth of the compartment adjacent to the substrate, as shown in Fig.4, Panel (a) in the main text.

**Appendix D:  Spin packet length**

Here, we elucidate the concept of spin packet length, $L_g$(Le Doussal and Sen (1992)). The spin packet length represents the characteristic length scale associated with molecular diffusion. As mentioned previously, the distance between the diffusion barriers, which defines the spatial range over which molecules can diffuse, is denoted by $L_s$. In NMR measurements, $L_s$ refers to the spatial extent to which PTFE molecules can move. However, in practical NMR relaxation measurements, the spatial region contributing to the magnetization is restricted, and not all nuclear spins within the sample are detected by NMR. In other words, in regions exposed to a strong magnetic field gradient, where magnetization relaxation is completed during the echo time before the signal is observed, such as near the diffusion barrier, the magnetization remains undetected even in regions where spins are present(Le Doussal and Sen (1992)).

Furthermore, even when considering a specific NAS in an MRI experiment, detecting NMR signals from all nuclear spins of interest within the NAS is impossible. Therefore, the effective diffusion barrier distance may be smaller than the physical distance. The spin packet length $L_g$ is the spatial region in which the NMR-active spins can be detected( Zhang and Hirasaki (2003)). Theoretical studies have indicated that this region, where spins are present but difficult to measure, is influenced by





variations in the magnetic field gradient experienced by the sample(Le Doussal and Sen (1992)). If the diffusion distance of the molecules is measured under varying magnetic field gradient strengths, the spin packet length $L_g$ may vary. However, in the experiment, the distance $d$, a parameter of the magnetic field gradient strength, was held constant between the tip of the needlelike ferromagnetic material and the sample surface, ensuring that the magnetic field gradient remained stable over time. Therefore, as the first approximation, the distance $L_s$ between the diffusion barriers and the observed spin-packet length $L_g$

can be considered to be proportional. Consequently, for the interpretation of $R_2$, we choose to use the spin packet length $L_g$ instead of $L_s$.

*Author contributions.* N.K. and N.A. designed the research and conducted the experiments. T.K. contributed to theoretical discussions. All authors wrote and reviewed the manuscript.

*Competing interests.* The authors declare no competing interests.

*Acknowledgements.* This work was partially supported by JSPS KAKENHI (22K05225 25K08761), Network Joint Research Center for Materials and Devices (#20241302, #20251279) and the Gunma University S-Membrane Project. One of the authors (NK) was support by JST SPRING, Grant Numver JPMJSP2146.





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
