# Peer review of "Static-Gradient NMR imaging for Depth-Resolved Molecular Diffusion in Amorphous Regions in Semicrystalline PTFE Film"

_Magnetic Resonance, 2025_

## Author Response (AR1)

**comments from referees:**

Upon reading the manuscript while unfamiliar with the authors's prior work, it took quite a bit of reading to understand what is meant by "substrate interface". Perhaps the authors could mention in the abstract (or early in the introduction) ".... of a poytetrafluoroethylene (PTFE) film immobilized on a glass substrate."

**author's response:**

We thank Reviewer #2 for highlighting the ambiguity in the term "substrate interface." In the present study, the PTFE film is affixed to a glass substrate using an epoxy resin layer; therefore, "substrate interface" specifically denotes the PTFE/epoxy-resin interface.

**author's changes in manuscriptp:**

We rewrote the relevant sentences in the following, P.1.L.5: "The results reveal that molecular mobility at the substrate interface of PTFE film immobilized on a glass substrate using epoxy resin is significantly constrained, ..."

P.6.L.131:"Conversely, near the glass substrate interface via epoxy resin, ..."

P.9.L.177: "We now focus on the region near the interface between the PTFE film and the glass substrate via epoxy resin."

**comments from referees:**

P 1, line 20. What about a reference to P.T. Callaghan, Translational Motion and Magnetic Resonance, Oxford University Press, Oxford, 2011?

**author's response:**

We appreciate Reviewer #2's suggestion. The previously cited 1984 article by P. T. Callaghan has been replaced with a citation to the recommended monograph.

**author's changes in manuscriptp:**

P.17.L.366:"Callaghan, P. T.: Translational dynamics and magnetic resonance: principles of pulsed gradient spin echo NMR, Oxford University Press, 2011."

**comments from referees:**

What is the number-averaged degree of polymerization? I have trouble visualizing the translational motion of the molecules. Is it reptation of linear polymer chains? What distances do the chain segments cover? How does the diffusive distance compare to the average size of the amorphous domains? Does the film have a lamellar morphology or is its morphology better described by the para-crystallinity model? Can you relate your findings to the chain motion in solid polyethylene (K. Schmidt-Rohr, H.W. Spiess, Macromolecules 1991) or to the free volume in the amorphous domains? How has the film been prepared?

**author's response:**

Because PTFE exhibits an ultrahigh melt viscosity, conventional methods for molecular-weight determination such as melt rheology or gel-permeation chromatography are not applicable. Consequently, we did not measure the molecular weight of our samples and used the commercial PTFE film as received. Industrial PTFE is generally known to possess molecular weights on the order of several million. Although the surface morphology of semicrystalline PTFE is not yet fully established, AFM studies (e.g., V. Korolkov, NANOscientific Magazine, 21 (2021)) report a multiscale hierarchical surface structure. At the 100 um scale, large domains ( 20 um) interconnected by rope-like features are observed. At higher spatial resolution ( 100 nm), highly oriented crystalline regions presumably lamellae on the order of 100 nm are separated by amorphous regions of several tens of nanometers. In our study, the characteristic distance between diffusion barriers, $L_s$, is therefore presumed to be on the order of several tens of nanometers. However, our technique does not allow

direct determination of $L_s$ or the effective spin packet length $L_g$, because the effective magnetic-field gradient contains a significant but unknown local component arising from the intrinsic magnetic-susceptibility variations of the PTFE sample.

We appreciate Reviewer #2's suggestion regarding comparisons with polyethylene; however, for brevity and clarity, we restrict the discussion to PTFE.

**author's changes in manuscriptp:**
P.13.L.271:"Although the surface morphology of semicrystalline PTFE is not yet fully established, AFM study has reported a multiscale hierarchical surface structure. At the 100 um scale, large domains ( 20 um) interconnected by rope-like features are observed. At higher spatial resolution ( 100 nm), highly oriented crystalline regions presumably lamellae on the order of 100 nm are separated by amorphous regions of several tens of nanometers. In our study, the characteristic distance between diffusion barriers, $L_s$, is therefore presumed to be on the order of several tens of nanometers. However, our technique does not allow direct determination of $L_s$ or the effective spin packet length $L_g$, because the effective magnetic-field gradient contains a significant but unknown local component arising from the intrinsic magnetic-susceptibility variations of the PTFE sample."

**comments from referees:**

Figure 1. It would help to identify BNd and Be in the figure. In the caption the second sentence could be rewritten as "The shape of the excited volume, determined by the ...".

**author's response:**
In accordance with Reviewer #2's suggestion, we have revised Figure 1 and its caption to explicitly identify $B_{Nd}$ and $B_e$, and updated the second sentence accordingly.

**author's changes in manuscript:**
The caption of Fig.1: "Figure 1. In our one-dimensional MRI approach, the resonance frequency was fixed and a spatially selective RF pulse with an RF field strength of nominal 50 kHz were applied. This pulse excites nuclear spins only within a narrow slice (the NMR-active slice;NAS), whose spatial position is defined by the static field gradient at the chosen resonance frequency. By incrementally varying the static magnetic field, the effective field gradient experienced by the sample changes, and consequently, the position of the NAS is shifted along the thickness direction of the film. The combined effect of the magnetic field generated by the needle-like ferromagnet, $B_{Nd}$, and the static magnetic field from the electromagnet, $B_e$ gives rise to tha mechanism of movement of NMR active slice."

**comments from referees:**

Figure 2. The labels b) and c) do not match the caption.

**author's response:**
We thank Reviewer #2 for pointing this out. The labels for panels (b) and (c) in Figure 2 have been corrected.

**author's changes in manuscript:**
We redrew Figure 2.

**comments from referees:**

P 7, line 132. Reference is made to the z-axis of a two-dimensional xy plot in Fig. 4a. Would it not be simpler to write "The contribution of diffusion to the spin-spin relaxation rate $R_2$ is color coded?"

**author's response:**
We appreciate the reviewer's suggestion and have revised the caption accordingly:

**author's change in manuscript:**
In the caption of Figure 4, we rewrote the relevant sentence in the follwoing. "... The contribution of diffusion to the spin-spin relaxation rate $R_2$ is color-coded (Panel (a)), where the relaxation exponent of the CPMG echo intensity $M(2n\tau)$ (for $n = 1$) is plotted as a function of the dimensionless diffusion coefficient (or dimensionless echo time) and position within the sample. ..."

**comments from referees:**

P 7, line 143. "This behavior of $R_2$ arises from its intrinsic modification instead of the diffusion process." What do you mean by intrinsic modification of the polymer film? Do you mean differences in polymer morphology between surface and film interior?

**author's response:**
We apologize for the ambiguous wording in the original manuscript. The reviewer's interpretation is correct: the observed difference in $R_2$ between the surface and the interior of the film originates from intrinsic variations in the local polymer morphology rather than from differences in diffusion.
We added the following sentence at the related paragraph.

**author's change in manuscript:**
P.7.L151:"The observed difference in $R_2$ between the surface and the interior of the film originates from intrinsic variations in the local polymer morphology rather than from differences in diffusion."

**comments from referees:**

P 9, line 171. "This behavior can be attributed to the system transitioning between the localization and motional averaging regimes, triggered by the change in tau as shown in Fig. 4(a).": How can the system (I assume you mean the polymer film) react to a pulse-sequence parameter like tau? A better way of writing may be "The observed behavior can be attributed to a transition of the observable from the localization regime to the averaging regime with increasing tau".

**author's response:**
We are grateful for this suggestion and have revised the manuscript accordingly.

**author's change in manuscript:**
P.9.L.179:"The observed behavior can be attributed to a transition of the observable from the localization regime to the averaging regime with increasing tau."

**comments from referees:**

P 9, line 84. Reference is made to Fig. 4c. I cannot find Fig. 4c.

**author's response:**
We thank Reviewer #2 for noting this inconsistency. The reference to Fig. 4c corresponds to the inset of the tau = 20 us data in Fig. 3.

**author's change in manuscript:**
L.9.L.193:"Specifically, when $\tau$ was set to 20 $\mu$s, and a regime transition was observed in the depth direction of the polymer film, as depicted in the plot for $\tau = 20$ $\mu$s in Figure 3."

**comments from referees:**

"Concerning the CPMG parameters, it would be helpful to state not only the values of tau (is that the echo time or half the echo time?) in the manuscript but also the pulse width."

**author's response:**

We thank the reviewer for this helpful suggestion. We have revised the second paragraph of Results and Discussion section to clearly state that tau denotes the delay between the 90 degree pulse and the subsequent 180 degree pulse, i.e., half of the echo time in the standard CPMG sequence. We have also added the RF intensity of the nominal 90 degree pulses used in our measurements.

**author's change in manuscript:**

We added the information about RF intensity we used at P.3.L.90, P.4.L.115, and caption of Fig.1.

**comments from referees (summary):**

"The manuscript seems to understate the influence of the RF field on selective excitation. In a grossly inhomogeneous field, every finite-length RF pulse is selective. How does your method obtain spatial resolution if not through selective excitation and detection?" "Essentially you are moving a selective volume along an extended object. The same situation is encountered in well-logging NMR albeit on different scales. This type of 1D imaging works because of the inherent combination of selectivity and movement of the selective volume, and one is not more important than the other. I think your revised paragraph does not get to the point."

**author's response:**

We thank the reviewer for this insightful comment. We agree that our one-dimensional MRI method relies on the combined effects of (i) spatially selective excitation and (ii) controlled displacement of the selective excitation volume, both of which are essential for obtaining depth-resolved information from the film sample. Following the reviewer's suggestion, we have substantially revised the relevant paragraph in the manuscript to clearly articulate this point.

In our method, the carrier frequency of the RF pulse is fixed, and a spatially selective excitation is generated using an RF amplitude of nominal 50 kHz under a static magnetic-field gradient. Because the excitation frequency is constant, the spatial position of the NMR-active slice is uniquely determined by the local magnetic field. By incrementally varying the static magnetic field, we modify the effective magnetic-field gradient experienced by the sample. This results in a shift of the spatial position that corresponds to the fixed excitation frequency, so that the NMR-active slice is translated along the film-thickness direction. Repeating this process allows us to sweep the selective volume across the entire film thickness and reconstruct a one-dimensional MRI profile.

We have clarified in the revised manuscript (the first paragraph in Section 2.1 Principle of depth profiling) that this mechanism indeed parallels the principle used in well-logging NMR, where the sensitive volume is translated through the medium by altering the magnetic-field configuration. We believe this revision directly addresses the reviewer's concern and makes the imaging principle more transparent to readers.

**author's change in manuscript:**

P.3.L.87:"In our one-dimensional MRI approach, we fix the resonance frequency and apply a spatially selective RF pulse with an RF field strength of nominal 50 kHz. This pulse excites nuclear spins only within a narrow slice (the NMR-active slice), whose spatial position is defined by the static field gradient at the chosen resonance frequency. By incrementally varying the static magnetic field, the effective field gradient experienced by the sample changes, and consequently, the position of the NMR-active slice is shifted along the thickness direction of the film. This procedure enables one-dimensional imaging along the sample depth without the need for frequency-swept selective excitation or gradient switching during RF irradiation. A similar principle–distance encoding realized by the translation of a spatially selective sensitive volume–has long been employed in well-logging NMR(¥citehurlimann2000spin). Our method shares the

same fundamental mechanism: the imaging contrast arises from the interplay between a localized excitation region and its systematic displacement through the sample. This conceptual parallel underscores the validity of our approach and places it within the broader class of 1D imaging methods that rely on spatial selectivity and controlled volume movement."
* * *
**comments from referees:**

Although years ago many authors, including myself, used the abbreviation PFG NMR for NMR diffusion measurements, it should be noted that pulsed field gradients could be merely being used for coherence selection. Consequently, it is better to use the expression pulsed gradient spin echo (PGSE) NMR, since a spin echo is necessarily sensitive to diffusion.

**author's response:**
We would like to thank the reviewer for the comment. We revised our manuscript in line with the comment.

**author's change in manuscript:**
P.1.L.19, L.21,P.2.L.33, L.43:The term PFG in the original manuscript was replaced by PGSE, which stands for "pulsed gradient spin-echo."
* * *
**comments from referees:**

The last paragraph on page 1 talks about the issues with using PFG (i.e., PGSE) NMR to study solids and the first paragraph on page 2 discusses the various static field gradient (SFG) methodologies available. Then somewhat out of place, the next paragraph starts by stating that despite the advances in PFG methods they are still not suited for solid-state applications. The contents of these chapters need to be reordered.

**author's response:**
We appreciate the reviewer's comment on the readability of Introduction. We rewrote the Introduction part to gain the readability.

**author's change in manuscript:**
We improved the logical flow by moving the paragraph that originally began with 'Despite these advances, ⋯' to precede the paragraph starting with 'To overcome these challenges, ⋯'. In conjunction with this relocation, we removed redundancies that emerged as a result of the reordering, and we added a transition word at the beginning of the revised paragraphs.

P.1.L.23:"... These limitations underscore the need for alternative approaches that are specifically tailored for solid materials.

Furthermore, electrically conductive solid components within the sample can generate substantial eddy currents, potentially degrading the specimen. Because eddy currents distort the NMR signal, it is necessary to wait an adequate period for their decay prior to acquiring a reliable measurement (Chapman et al. (1957); Gibbs and Johnson Jr (1991); Price (1998)). In addition, conventional high-frequency NMR systems that rely on superconducting magnets demand extensive operational infrastructure- large-scale facilities, cryogenic cooling, and vacuum environments–which imposes significant financial and logistical burdens. Thus, there remains a pressing need for a simple, cost-effective, and versatile MRI method that can provide spatially resolved information on molecular dynamics in solid systems.

To overcome these challenges, ..."

Page 2, line 40. "Additionally, the abrupt shifts ⋯ can lead to prolonged dead times in the magnetic resonance signal ⋯". This seems confused – at least in how it is written. It is not that there is no signal, it is that the signal is scrambled by eddy currents and other effects and so one needs to wait a sufficient amount of time for these to decay before you can acquire a useful signal.

**author's response:**
We thank the reviewer again for their comment. We replace the sentence in original manuscript to the following.

**author's change in manuscript:**
P.2.L.26:"Because eddy currents distort the NMR signal, it is necessary to wait an adequate period for their decay prior to acquiring a reliable measurement."

Page 2, line 69. CPMG should have been defined here (i.e., the first occurrence) and not on Page 5 line 105 or Page 14 line 282 with the references (move these to page 2 as well).

**author's response:**
We would like to thank the reviewer again. We revised our manuscript in line with the comment.

**author's change in manuscript:**
P.3.L.64:"To complement our previous work, we examined the influence of translational diffusion on spin-spin relaxation using an $R_2$-dispersion approach (Yu (1993)) by systematically varying the echo time in the Carr-Purcell Meiboom-Gill(CPMG) sequence (Carr and Purcell (1954); Meiboom and Gill (1958)). The CPMG sequence was employed for the MRI measurements, and the dependence of $R_2$ on translational diffusion was examined by varying the half echo time, $\tau$, as follows: Depth-resolved one-dimensional imaging was achieved by stepwise modulation of the static magnetic field strength using a normal-conducting electromagnet. The RF intensity for the CPMG method was nominal 50 kHz, which was calibrated using CPMG pulse se- quence under homogeneous resonant magnetic field only from the electromagnet. At each magnetic field point, 256 signal accumulations were acquired at a resonance frequency of 29.750000 MHz. The decay plots and fitting curves for the CPMG measurements are shown in Appendix A. Furthermore, the influence of the intrinsic $R_2$ on the experimentally obtained $R_2$ was negligibly small(see Appendix B). However, in the $R_2$-dispersion method employed in this study, accurately determining the diffusion coefficient is challenging due to the ambiguous nature of the local magnetic field gradient within the sample. Consequently, it should be emphasized that the diffusion analysis presented here is qualitative in nature."

Page 3, line 81. I am a little mystified by comment "reaches the required threshold for the onset of nuclear magnetic resonance". My point is that everywhere is resonating to some degree (e.g., the Earth's Magnetic field is about 50 μT). Do the authors mean resonating in a particular frequency range?

**author's response:**
Thank you for drawing our attention to this important point. Our intention was to refer specifically to a particular range of resonance frequencies. We have completely rewritten the relevant paragraph, and we believe this revision resolves the reviewer's concerns.

**author's chenge in manuscript:**
This question is relevant to the above question concerning the selective excitation by RF pulses and we have completely rewritten the relevant paragraph(P.3.L.87).

**comments from referees:**

Page 4, Fig. 1. Please define "NAS".

**author's response:**

We would like to thank the reviewer for their comment. We have completely rewritten the caption of Fig. 1. In doing so, we explicitly clarified the original term behind the abbreviation "NAS," namely "NMR-active slice."

**author's change in manuscript:** This question is relevent to the above question concerning Figure 1 and the revised caption is shown above.

**comments from referees:**

Page 7, line 132. "···diffusion is not in progress···"? Do you mean "diffusion is not measurable because of the short $\tau$"? But diffusion is always occurring.

**author's response:**

We did not mean to suggest that diffusion cannot be measured because $\tau$ is short; rather, our intention was to indicate that when $\tau$ is short, the spins have not yet reached the diffusion barrier and therefore undergo free diffusion.

**author's change in manuscript:**

P.7.L.140:"When the echo time $\tau$ is short, and the spins have not yet reached the diffusion barrier and therefore undergo free diffusion. Therefore, spins remain within the short-time regime."

**comments from referees:**

Page 11, line 232. Please give a reference for the "Landweber iteration method".

**author's response:**

We added four references for the Landweber iteration method.

**author's changes in manuscript:**

P.11.L.239:"While this characteristic constitutes a limitation of the present method, it also suggests the possibility that, in the future, inverse problem approaches to convolution–such as the Landweber iteration method (Degen et al. (2009); Chao et al. (2004); Landweber (1951); Spencer and Bi (2020)) –may enable imaging with significantly enhanced spatial resolution."

references.

Degen, C., Poggio, M., Mamin, H., Rettner, C., and Rugar, D.: Nanoscale magnetic resonance imaging, Proceedings of the National Academy of Sciences, 106, 1313–1317, 2009.

Chao, S.-h., Dougherty, W. M., Garbini, J. L., and Sidles, J. A.: Nanometer-scale magnetic resonance imaging, Review of Scientific Instru ments, 75, 1175–1181, 2004.

Landweber, L.: An iteration formula for Fredholm integral equations of the first kind, American journal of mathematics, 73, 615–624, 1951.

Spencer, R. G. and Bi, C.: A tutorial introduction to inverse problems in magnetic resonance, NMR in Biomedicine, 33, e4315, 2020.

**Typographical Corrections**

**comments from referees:**

Many references run against the previous word (e.g., page 1, line 11 "viscoelasticity(Keddie et al. ")). This type of error is throughout the manuscript.

**author's response:**
We revised our manuscript in line with the comment.

**author's change in manuscript:**
We added "space" before all the citations in the revised manuscript.

**comments from referees:**

There is a mysterious "8" in the middle of the text on line 34 on page 2.

**author's response:**
We omitted the mysterious "8".

**author's change in manuscript:**
P.2.L.40: We omitted the mysterious "8".

**comments from referees:**

"NMR" was defined on page 1, there is no need to redefine it on page 3 line 83. Indeed, given the journal that this manuscript has been submitted to it is doubtful that NMR needs to be defined at all.

**author's response:**
We revised our manuscript in accordance with the comment, except for the first occurrence of the term "NMR."

**author's change in manuscript:**
This question is relevent to the above question concerning selective excitation and during the revision of the manuscript the relevent sentence has been removed.

**comments from referees:**

Page 6, line 177. "$\cdots$ no variations present in R2 near $\cdots$" sounds odd. How about "$\cdots$ no variations in R2 are present near $\cdots$"

**author's response:**
We revised our manuscript in line with the reviewer's suggestion.

**author's change in manuscript:**
P.6.L.124:"As shown in Fig.3, no variations in $R_2$ are present near the air-side surface of the film, ($B_e$ ¡0.762 T) for different $\tau$ values."

**comments from referees:**

Page 8, Fig. 4. Some of the variables in the axes labels should be italicised.

**author's response:**
We redrew Fig.4 in line with the reviewer's suggestion.

**author's change in manuscript:**
We redrew Figure 4.

**comments from referees:**

Page 9, line 3. Instead of " when τ = 20 μ s was held constant ", did the authors mean " when τ was set to 20 μ s " ?

**author's response:**
The reviewer's comment is correct. We revised our manuscript in line with the reviewer's suggestion.

**author's change in manuscript:**
P.9.L.191:"Specifically, when $\tau$ was set to 20 $\mu$s, and a regime transition was observed in the depth direction of the polymer film, as depicted in the plot for $\tau$= 20 $\mu$s in Figure 3."

**comments from referees:**

Page 10, line 194. It is a small point and I know that " rotational motion " is in common usage (so there is no need to change), but " rotational " implies rotating about one axis, whereas the molecules are tumbling and so to me " reorientational " seems more appropriate.

**author's response:**
We would like to thank the reviewer for their helpful comment. We have revised the text by adding " reorientation " alongside " rotational. "

**author's change in manuscript:**
P.3.L.62:"This discrepancy may arise because $R_1$ reflects only rotational (more precisely, reorientational) molecular dynamics and is insensitive to translational diffusion."

**comments from referees:**

Page 13, line 247. Boembergen $\rightarrow$ Bloembergen

**author's response:**
We would like to thank again for the comment. We corrected the missspelling of "Bloembergen."

**author's change in manuscript:**
P.13.L.255:"According to the Solomon-Bloembergen theory (Solomon (1955)), ..."